# Tailored Treatment Options for Cerebral Cavernous Malformations

**DOI:** 10.3390/jpm12050831

**Published:** 2022-05-20

**Authors:** Jessa E. Hoffman, Blake Wittenberg, Brent Morel, Zach Folzenlogen, David Case, Christopher Roark, Samy Youssef, Joshua Seinfeld

**Affiliations:** Department of Neurosurgery, University of Colorado, Aurora, CO 80045, USA; blake.wittenberg@ucdenver.edu (B.W.); brent.morel@ucdenver.edu (B.M.); zach.folzenlogen@ucdenver.edu (Z.F.); david.case@ucdenver.edu (D.C.); christopher.roark@ucdenver.edu (C.R.); samy.youssef@ucdenver.edu (S.Y.)

**Keywords:** cerebral cavernous malformations, CCMs, cavernomas

## Abstract

The diagnosis and treatment of cerebral cavernous malformations (CCMs), or cavernomas, continues to evolve as more data and treatment modalities become available. Intervention is necessary when a lesion causes symptomatic neurologic deficits, seizures, or has high risk of continued hemorrhage. Future medical treatment directions may specifically target the pathogenesis of these lesions. This review highlights the importance of individualized treatment plans based on specific CCM characteristics.

## 1. Introduction

The diagnosis and treatment of cerebral cavernous malformations (CCMs), or cavernomas, continues to evolve as more data and treatment modalities become available. Cerebral cavernous malformations (CCMs) are hamartomas with endothelium-lined vascular chambers [1,2]. CCMs lack developed vascular structures like elastic lamina, smooth muscle, or tight junctions; have no interspersed brain parenchyma; and have a border of nonfunctioning glial tissue [1,3].

CCMs are considered benign lesions and intervention is generally not required if asymptomatic [2,4]. However, CCMs can cause major neurologic deficits and significantly affect the quality of a patient’s life when symptomatic [2]. CCMs can lead to symptoms by causing uncontrolled seizures or through bleeding events, leading to focal neurologic deficits. In this review, we discuss the etiology of CCMs and current management strategies available depending on the characteristics of individual cases in order to guide diagnosis and treatment discussions with patients.

## 2. Epidemiology and Genetics of CCMs

CCMs are rare lesions found in 0.4–0.8% of the population [1,5]. They are found to be present at a rate up to 1.5% in the Hispanic population [6]. Between 40% and 60% of cavernomas are familial and these patients are more likely to have multiple lesions [2] (Figure 1A,B). There have been three distinct genes identified that are associated with familial cavernomas. These include loss-of-function mutations at CCM1/KRIT1, CCM2/MGC4607, and CCM3/PDCD10 which are thought to contribute to an immature vascular complex [7]. The protein complex found in CCMs is proposed to have impaired inter-endothelial tight junctions and hyperpermeability with altered vasculogenesis leading to formation of the lesion [7]. As opposed to CCMs that occur sporadically, patients with familial CCMs have been found to develop new lesions over their lifetime.

## 3. Diagnosis

CCMs are identified by contrasted computed tomography (CT) or magnetic resonance imaging (MRI). As they are low flow lesions by nature, CCMs are not visible on digital subtraction angiography (DSA). For this reason, MRI is the diagnostic modality of choice [1,2]. CCMs have a characteristic appearance of vascular channeling with tangles of flow voids on noncontrasted T2 weighted sequences which can be helpful for diagnosis. Recent use of gradient-echo sequencing facilitates identification of lesions based on hemosiderin deposits within and around CCMs (Figure 1B). However, use of these sequences alone does exaggerate the size of the lesion since hemosiderin staining can occur in the parenchyma adjacent to the lesion itself [2]. This hemosiderin leakage may be due to inherent vascular permeability which may be a key to the pathogenesis of CCMs themselves [7].

## 4. Natural History and Rate of Hemorrhage

A large prospectively controlled study has not been carried out to determine the definitive risk of hemorrhage of CCMs. Bleeding risk is complicated by factors such as location, genetics, and prior hemorrhage, all of which contribute to each lesion’s future risk of bleeding over time. Overall, CCMs are found to have a 0.8–3.8% risk of bleeding per year [8,9]. The risk of hemorrhage per year increases drastically to 7.0–8.9% in the period immediately following a hemorrhage [2,4]. The risk of hemorrhage, especially asymptomatic, may be slightly higher in familial cases of CCM which have a reported rate of symptomatic hemorrhage of 1.1% per lesion year and an asymptomatic hemorrhage rate of 13% per patient-year [10].

The incidence of symptomatic hemorrhage and clinical presentation appears to vary significantly depending on the location of CCMs. Supratentorial cavernomas make up 65–80% of CCM cases and have a bleeding rate of 0.4% per year [8,11,12]. Supratentorial CCMs, if symptomatic, commonly present with headaches or seizures [1,13,14]. Brainstem CCMs are more likely to present with neurologic deficits after hemorrhage [2,8]. The risk of clinically significant bleeding from infratentorial CCMs ranges from 2.46–3.8% per person-year [8]. This risk is even higher after an initial bleeding event with a risk of up to 21% per year [15]. However, some data suggest overall hemorrhage rates as low as 0.05% per patient-year after an initial bleeding event [16]. The relatively high rates of brainstem CCM hemorrhage may simply reflect a higher likelihood of identification of bleeding events since a hemorrhagic event in the brainstem is likely to be symptomatic due to the eloquence of the surrounding brainstem tissue [2]. Regardless, brainstem CCMs are more likely to cause neurologic deficit or death when hemorrhagic.

It is clear that individual CCMs are more likely to bleed if they have bled before [2,4]. As mentioned, there is an overall hemorrhage risk of up to 8.9% per year after a prior bleed [4]. However, this increased risk is proposed to be limited to a discrete period of time, after which the risk of hemorrhage may naturally return to the baseline level of risk [2,17]. Other factors that increase the risk of bleeding include association with a developmental venous anomaly [1,18,19]. Young age and large size at presentation were found to increase the risk of hemorrhage per year as well [1,20]. Additionally, some studies have noted a trend towards an increased risk of hemorrhage in females, although this has not been robustly proven [21].

## 5. Management of Incidental Cavernomas

Up to 44% of CCMs are asymptomatic [1,6]. Asymptomatic supratentorial cavernomas can be observed over time, especially if there is no evidence of hemorrhage (Figure 1). Recent guidelines recommend asymptomatic lesions do not require any intervention unless they become symptomatic (Figure 2) [1,2]. Brainstem CCMs that are found incidentally should be watched closely as they have a potentially higher rate of hemorrhage, as well as higher morbidity in case of bleeding [8]. However, the risk of surgical intervention is usually warranted only after a brainstem CCM has had one or more hemorrhagic events.

## 6. Management of Symptomatic Supratentorial Cavernomas

Surgical intervention is the first treatment option for accessible supratentorial lesions causing uncontrolled seizures or symptomatic lesions due to recurrent hemorrhage or mass effect (Figure 3). Resection mitigates any further risk of hemorrhage and permanent neurologic deficits in addition to seizure improvement in 80% of cases [13,22]. Supratentorial lesions that are cortically-based, supratentorial, symptomatic, or have bled are favorable for resection and have a high likelihood of successful symptomatic improvement and seizure control [1,22,23]. Surgical resection is especially beneficial in patients whose seizure-onset is acute and caused specifically by the lesion [22,24]. Symptoms due to mass effect from acute hemorrhage noted on neurological examination are likely to improve over time regardless of treatment modality, but lesions that present with recurrent hemorrhage should be resected to prevent progressive and permanent neurologic deficits [1,2]. This must be taken in the context of the cumulative risk of hemorrhage over time, which is higher in the immediate post-hemorrhage period but returns to baseline after 2–3 years [17].

## 7. Management of Symptomatic Infratentorial Cavernomas

Brainstem lesions should be surgically resected if they demonstrate recurrent bleeding or progressive neurologic deficits (Figure 4 and Figure 5). The risk of new neurologic deficit is significant in these lesions and can be devastating, or even fatal [1,21]. However, surgical intervention in the acute hemorrhagic period is not advised (Figure 5A). Maturation of the lesion promotes the development of a distinct plane between important brainstem tissue and the lesion itself. (Figure 5B). This allows for the safest approach to brainstem cavernoma surgical resection and prevents additional neurologic morbidity. Minimizing surgical complications with resection of brainstem CCMs is crucial and difficult as the rate of immediate postoperative deficit may be as high as 67%, with at least a 15% perioperative risk of permanent moderate to severe disability or death [15,21,25,26]. Despite the risks associated with resection, surgery remains the best option in select cases of symptomatic brainstem CCMs given the high risk of further permanent disability and death with recurrent hemorrhagic events without surgery (Figure 5) [1,2,15,27]. If observed after an initial hemorrhage, surgery should be re-considered if recurrent hemorrhage or progressive neurologic decline occurs [1,21]. This is especially true if a deeply located CCM poses an excessive risk for surgical resection at presentation but subsequently expands to the surface from recurrent hemorrhage and becomes more surgically accessible.

Hemorrhagic cerebellar CCMs may also have devastating consequences, given their confinement in the posterior fossa where a sizable hematoma can result in brainstem compression and obstructive hydrocephalus. Fortunately, the location of a cerebellar CCM often lends itself to surgical resection with a low risk of serious neurologic morbidity. For this reason, cerebellar CCMs can be considered for resection after a single hemorrhagic event if the risk of recurrent hemorrhage is considered significant and expected surgical morbidity is acceptable [1,28].

## 8. Stereotactic Radiosurgery in The Treatment of CCMs

Stereotactic radiation remains a controversial topic for CCMs that carry unacceptable surgical risk, such as brainstem CCMs that do not come to the surface or lesions in exquisitely eloquent brain tissue. Stereotactic Radiosurgery (SRS) has been considered an option as an extrapolation from the success shown by SRS treatment for arteriovenous malformations (AVM) [29]. However, differences including pathogenesis, etiology, growth, and blood flow may render SRS less effective in CCMs.

Overall, SRS has been shown to decrease rates of hemorrhage after a period of 2 years [27]. However, during the initial post-treatment period, the treated lesion shows a higher risk of hemorrhage [2,29]. Additionally, this temporal clustering of hemorrhage with a later return to baseline risk may simply mirror the natural history of cavernous malformations [17]. SRS carries its own risks, including hemorrhage, edema, and increased seizure frequency, that approach the operative morbidity of high-risk surgical lesions [29]. Data on SRS may be complicated by selection bias since lesions treated with SRS were not selected for surgery initially due to their inherent high risk. As a result, further high-quality investigations are necessary to define the true benefit and limitations of SRS. However, for now SRS does remain an option for symptomatic lesions that are not candidates for surgical resection.

## 9. Pharmacotherapy

Seizures are a common presentation of CCMs as a result of cortical irritation [30]. While not aimed at eliminating or reducing the size of CCMs, antiepileptic medications, such as levetiracetam or lacosamide, are a good initial treatment for seizures caused by CCMs. Adequate seizure control can be achieved in as many as 60% of patients with medication alone [30]. Surgical intervention is often an excellent option if seizures continue despite medical treatment, and for patients who do not want to be on lifelong seizure medications as up to 80% of patients achieve seizure control following resection [22,30].

There are a number of experimental pharmacologic treatments that aim to address the development and growth of CCMs. The proteins involved in familial CCM syndromes appear to be involved in the maturation of vasculogenesis, so therapy may be directed at promoting a mature vascular complex and creating functional interendothelial cell junctions [7].

Novel treatments for symptomatic cavernomas suggest a role for propranolol which may work by inhibiting new blood vessel growth [31,32]. Treatment of aggressive lesions with propranolol in select cases has led to partial regression and symptom alleviation; however, similar to SRS, it remains to be seen whether this is a reflection of the pharmacologic intervention or demonstration of the natural history of these lesions [31,33]. By this logic, further intervention may be possible with medications aimed at the general vascular development process to promote maturation and interendothelial junction formation.

One of the molecular pathways found to be intricately involved in both CCM development and vasculogenic maturation is the RhoA kinase pathway [7,34,35]. The RhoA protein may be increased in the genetic development of CCMs which leads to endothelial cell dysregulation. The inhibition of RhoA may be another target to prevent development or progression of CCM lesions [7,35]. One proposed treatment through the RhoA kinase pathway includes statin medications which may stabilize the permeability of CCM vascularity and decrease activation of RhoA kinase [7,31,34,35]. Statin therapy is a promising medication as it is already widely in use and has an acceptable side effect profile for many patients. Its use as a CCM therapy has shown it may decrease chronic hemorrhage and decrease vascular permeability; however the extent of benefit needs far more data to be considered as a standard therapy [7,36]. One study did not find significant decrease in vascular permeability as measured by dynamic contrast enhanced perfusion MRI for CCMs treated with statins, however this may reflect imaging results rather than true clinical usefulness [35].

Fasudil is another medication in trial that may address the pathogenesis of CCM lesions through the RhoA kinase pathway [36,37]. Fasudil is a specific RhoA kinase inhibitor which has shown promise in some models of genetic CCM development and stabilizing the interendothelial junctional complex. When tested against simvastatin, fasudil was found to be more effective in increasing survival and decreasing development of CCMs [36]. There are a number of additional manufactured drugs which target the RhoA kinase pathway for CCM and show promise in preclinical tests. However, none have made it to Phase III trial for CCMs at the time of this review.

Interestingly, use of antiplatelet and anticoagulation medications does not appear to increase rates of hemorrhage in patients with CCMs [38]. These medications have been found to be safe in the setting of CCMs when needed to treat unrelated comorbidities [36]. In fact, one study suggests there may even be some protection against hemorrhagic events with the use of these medications; however, the long-term results still remain to be seen as this study may not have [38]. The exciting progress leading to further elucidation of the molecular pathogenesis of CCMs will likely provide insight into useful pharmacotherapy that may prevent development and/or hemorrhage of CCMs [7,34,36].

## 10. Future Interventional Treatment Directions

To date, no targeted therapies focusing on the genetic mutations responsible for familial CCM syndromes are available. However, future genetic treatments could capitalize on the role of these genes to repair the vasculogenesis pathway and potentially prevent the development and progression of CCMs [7]. Gene therapy is proposed as a viable option to replace dysfunctional vascular proteins seen in the familial forms of CCMs [7]. Although not fully understood, a defect in the complex inter-endothelial protein interaction is thought to be involved in the formation of these lesions including the RhoA kinase pathway [7]. Novel pharmacologic and genetic therapies will likely promote the formation of a mature complex [7]. Genetic therapies replacing the dysfunctional CCM1, CCM2, and CCM3 genes would theoretically provide the framework to return people with this disease to a normal state [7]. Future research will hopefully continue to elucidate the etiology of these lesions with the goal of prevention of de novo CCMs and halting progression of existing lesions.

Additionally, laser ablation may be an option for nonresectable lesions in deep structures of the brain. Laser ablation would be available as a less invasive option than open surgical resection, but the consequences of ablation remain to be seen. The feasibility of this option was shown recently by MacCracken et al. with five prospectively chosen patients with intractable epilepsy secondary to cavernomas treated with laser ablation [39]. Their results demonstrated diminution of the lesions over time without perioperative complication and an 80% postoperative seizure control rate [39]. This series was limited to lobar CCMs, so it is unclear how results of this technique will translate to lesions in eloquent, deep supratentorial, infratentorial, or brainstem locations [39].

## 11. Conclusions

Cavernous malformation management must be tailored to each unique patient’s situation with consideration of the context of each lesion including potential hemorrhagic risk, operative morbidity, and conservative management options. In general, cavernous malformations that are asymptomatic do not require intervention. If symptoms or recurrent bleeding occurs, surgical resection can be curative for lesions with acceptable perioperative risk. Supratentorial lesions are more likely to cause seizures and can be resected surgically if seizures or hemorrhagic symptoms are not controlled with medication. Infratentorial lesions are more likely to have recurrent and devastating bleeding events. The risk of surgical resection may be warranted, especially in brainstem CCMs that demonstrate recurrent hemorrhage. There is some evidence for SRS as an option for non-operable lesions. However, it remains to be seen whether radiation is a curative treatment option for CCMs. Future directions in management of CCMs will likely include less invasive surgical techniques such as laser ablation as well as pharmacologic and genetic therapies aimed at repairing the underlying mechanisms responsible for CCM formation.

## Figures and Tables

**Figure 1 jpm-12-00831-f001:**
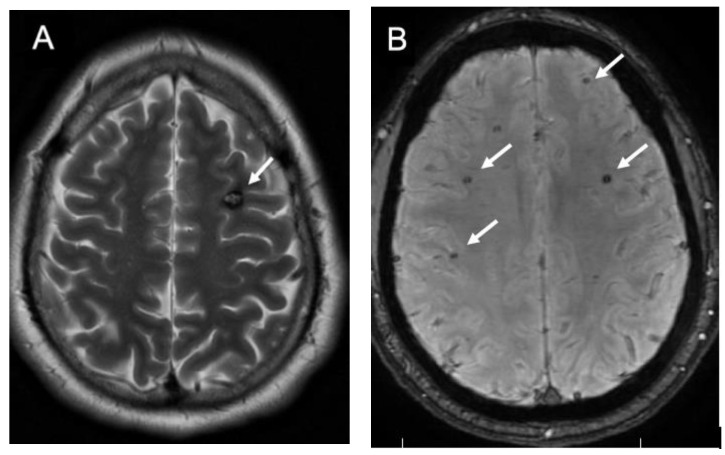
Nonoperative Asymptomatic CCMs. (**A**) patient with familial CCM and a left frontal lesion on T2 axial MRI ((**A**), white arrow). The patient has numerous additional lesions identified as hypointensities seen best on gradient-echo sequences (representative arrows, (**B**)). These lesions were not causing any symptoms and were monitored without intervention.

**Figure 2 jpm-12-00831-f002:**
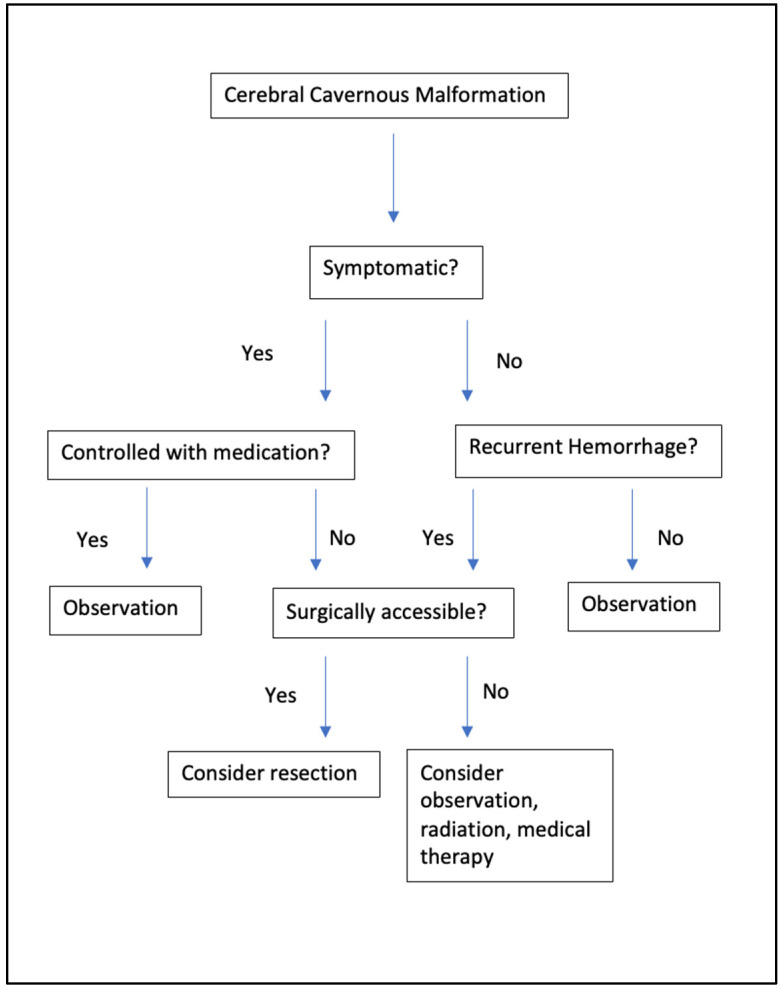
Management algorithm for CCMs.

**Figure 3 jpm-12-00831-f003:**
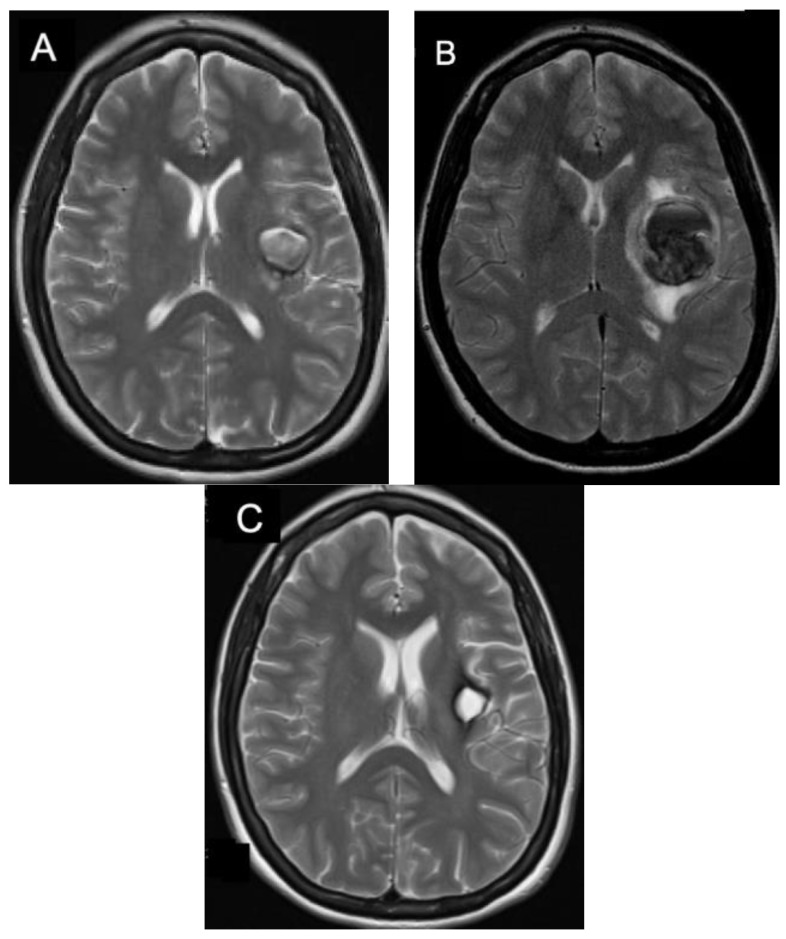
Operative Symptomatic Supratentorial CCM. T2 axial MRI of a patient with an enlarging hemorrhagic left insula CCM initially presenting with word finding difficulties and right sided weakness (**A**). The CCM was found to have enlarged over the next three months (**B**). The patient underwent surgical resection she experienced almost immediate resolution of her symptoms with T2 MRI imaging three months after surgery showing complete resection (**C**).

**Figure 4 jpm-12-00831-f004:**
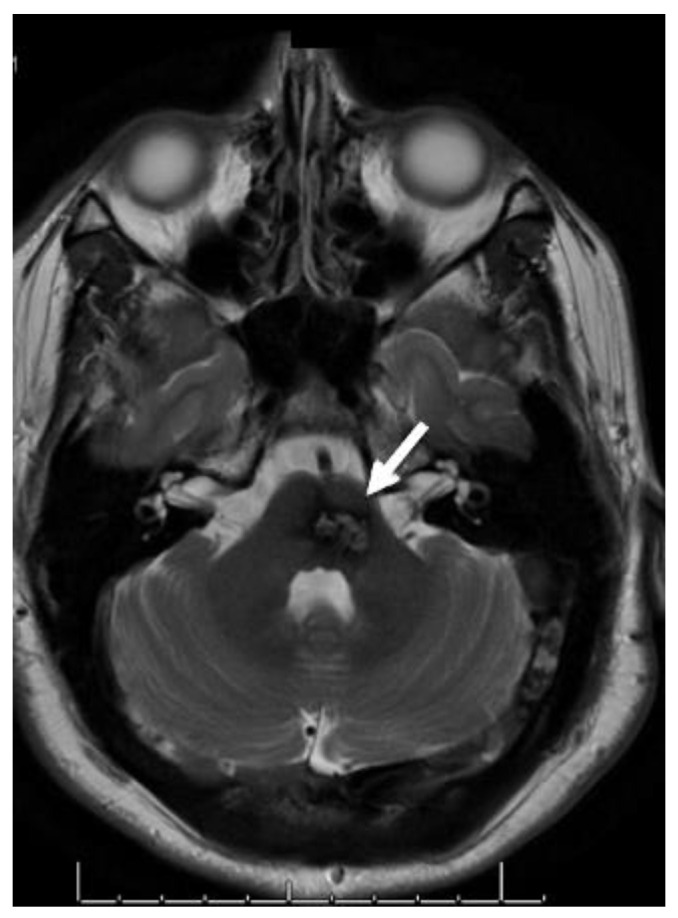
Non-operative Brainstem CCM. T2 axial MRI of a patient with familial CCMs found to have a non-hemorrhagic brainstem lesion (white arrow) that does not come to the surface. This patient was closely monitored without operative intervention.

**Figure 5 jpm-12-00831-f005:**
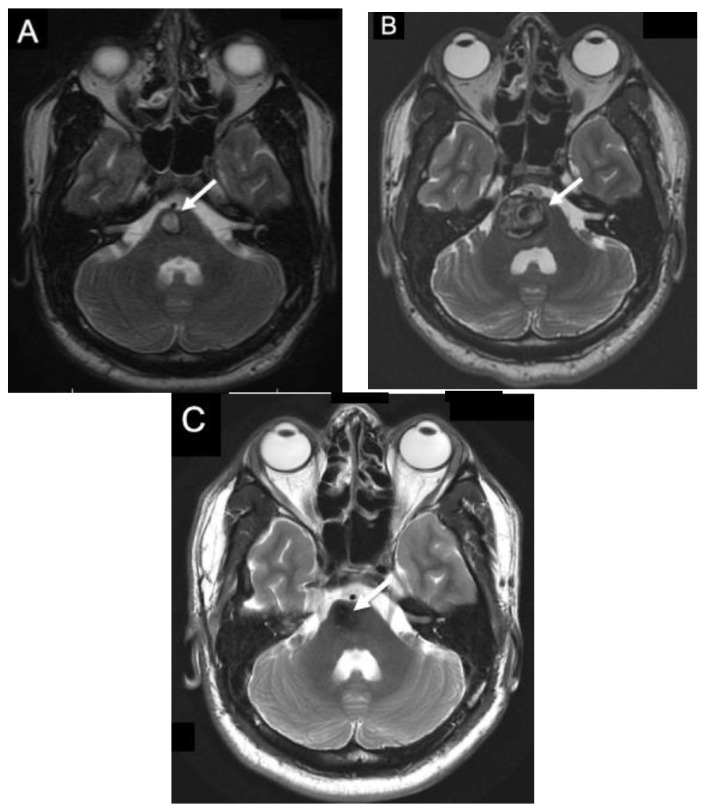
Operative Brainstem Lesion. T2 axial MRI of a 27-year-old male patient found initially to have a pontine CCM ((**A**), white arrow). He experienced two hemorrhagic events in a 6-month period resulting in left sided weakness, numbness, and slurred speech with enlargement and hemorrhage into the lesion seen on T2 MRI (**B**). He was taken for complete resection of the lesion one month after the second hemorrhagic event via a trans-petrosal approach with resolution of brainstem compression. The patient had progressive recovery over the following 6 months at which time a T2 MRI showed no residual CCM (**C**).

## Data Availability

Not applicable.

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
