# Peer review of "Tailored Treatment Options for Cerebral Cavernous Malformations"

_jpm, 2022, doi:10.3390/jpm12050831_

Round 1
Reviewer 1 Report
The authors reported a review paper on cerebral cavernous malformations focusing on management. The review is interesting but I have some major and minor concerns
Introduction: the final aim of the review and the utility for clinicians should be precised better.
Epidemiology and genetics: Epidemiology part should be widened with data on prevalence in different countries. Genetic subtypes should be detailed better since they have precise clinical characteristics depending on mutation. I would suggest to separate epidemiology in a single chapter and create a chapter of pathophisiology and genetics.
Diagnosis part should be improved. THis part is poor and does not include contrast enhancement, CTA, MRA or DSA for differentual diagnosis as well as modern diagnostic techniques. Site of lesions should be also reported. Association with caput medusae is not mentioned at all.
Management: a table resuming studies on therapy should be added to help the reader.
Figures should be resumed in a single panel.They are too many at present state
Author Response
Thank you for this comment. This has been clarified in the introduction.
As CCMs are an overall rare occurrence, we would like to leave this as a broad and accessible discussion of the general knowledge of CCMs. Unfortunately, the breadth of this manuscript does not delve into the intricacies of CCM genetic subtypes, but the references we listed do highlight those. As the pathophysiology remains unknown regardless of subtype I look forward to more detailed treatments as this is explored.
I hope you will find that the discussion of DSA and MRI for diagnosis are now addressed, as well as the current diagnostic technique gradient echo imaging. I believe these are the most useful for diagnosis. I have not found any suggestions that CTA or MRA is the most useful for diagnosis, however if this is the case I would be interested to see the references and reasoning. Hopefully our diagnosis section points providers in the most clear direction towards diagnosis. We do mention association with developmental venous abnormalities aka caput medusae in the natural history section.
Thank you for this comment, we have added a flow chart for management.
My understanding is that the figures will be paneled together in the final manuscript such that there will be figures with sub panels. I have formatted this in the manuscript to reflect this.
Thank you.
Reviewer 2 Report
Excellent overview of diagnosis, natural history, treatment options and outcomes for CCM. Beautiful illustrative images were included to enrich the mansucript. My only mind comment is to add a paragraph to pharmacotherapy mentioning the use of aspirin and oral anticoagulation, which counter-intuitively appears to reduce risk of hemorrhage.
Author Response
Thank you for this comment, we have added a discussion of antithrombotics which may decrease risk of hemorrhage. It appears as though this is still somewhat controversial and further data will be helpful for further conclusions to be made, but is certainly an interesting topic to explore going forward.
Reviewer 3 Report
Dear authors thank you for this article.
It would be nice to see the database analysis of the articles as it is an review article.
Please include an diagram of the treatment options.
Please put an/or arrow(s) on the figure 1B.
In the pharmacotherapy section it would be nice to include some AET therapy suggestions and eventually to discuss the neuromonitoring options in the acute phase.
I think that the article is useful and interesting.
Author Response
Thank you for this comment, I do not feel the database analysis adds much to the content of this article in the limitations of the figure and word count, however if it is felt to be essential it can be included.
Thank you for this comment, a diagram has been added as Figure 4.
Thank you, arrows have been added to figure 1B.
Thank you, we have added AED suggestions. We have clarified what we mean by monitoring as attention to neurologic examination.
Reviewer 4 Report
This is an interesting review article discussing several tailored treatment options for cerebral cavernous malformations. There have been additional drugs/nutritional supplements, in addition to propranolol, currently being used or in clinical trials such atorvastatin, rapamycin, tempol, etc. It would be nice to briefly discuss the underlying mechanisms on how propranolol and other potential drugs play a role in antagonizing the effects in CCM disease. Overall, the manuscript was written in good English and easy to follow and understand.
Author Response
Thank you very much for your commentary. The breadth of this review may not be enough to include and specify all potential pharmacotherapies in use and in trial. I have, however, added clarity on the potential mechanism of propranolol and the role of medical therapies with future directions and data. Thank you.